# Female Human Papillomavirus Infection Associated with Increased Risk of Infertility: A Nationwide Population-Based Cohort Study

**DOI:** 10.3390/ijerph17186505

**Published:** 2020-09-07

**Authors:** Li-Chuan Hsu, Kuan-Hao Tsui, James Cheng-Chung Wei, Hei-Tung Yip, Yao-Min Hung, Renin Chang

**Affiliations:** 1Department of Obstetrics and Gynecology, Kaohsiung Veterans General Hospital Pingtung Branch, Pingtung 91245, Taiwan; taylor6174@gmail.com; 2Department of Healthcare Administration and Medical Informatics, Kaohsiung Medical University, Kaohsiung 80708, Taiwan; 3Department of Obstetrics and Gynecology, Kaohsiung Veterans General Hospital, Kaohsiung 81362, Taiwan; khtsui60@gmail.com; 4Department of Pharmacy and Master Program, College of Pharmacy and Health Care, Tajen University, Pingtung 90741, Taiwan; 5Division of Allergy, Immunology and Rheumatology, Chung Shan Medical University Hospital, Taichung 40201, Taiwan; jccwei@gmail.com; 6Institute of Medicine, Chung Shan Medical University, Taichung 40201, Taiwan; 7Graduate Institute of Integrated Medicine, China Medical University, Taichung 40402, Taiwan; 8Management Office for Health Data, China Medical University Hospital, Taichung 404332, Taiwan; fionyip0i0@gmail.com; 9Department of Internal Medicine, Kaohsiung Municipal United Hospital, Kaohsiung 80457, Taiwan; 10Department of Medicine, School of Medicine, National Yang Ming University, Taipei 11221, Taiwan; 11Department of Senior Citizen Service Management, Yuh-Ing Junior College of Health Care and Management, Kaohsiung 80776, Taiwan; 12Department of Emergency Medicine, Kaohsiung Veterans General Hospital, Kaohsiung 81362, Taiwan; 13Department of Recreation Sports Management, Tajen University, Pingtung 90741, Taiwan

**Keywords:** human papillomavirus infection, infertility, cohort study

## Abstract

Objective. This study investigated whether women with a history of human papillomavirus (HPV) infection have an increased risk of infertility. Material and Methods. All patients with an HPV infection (n = 11,198) in Taiwan’s National Health Insurance Research Database (2000–2012) were propensity score matched with control subjects (n = 11,198) without an HPV infection by age, sex, index year, and relevant co-morbidities. Both groups were tracked until a diagnosis of infertility was recorded. The Chi-square test was used to analyze the distribution of demographic characteristics in the HPV group and non-HPV group. A Cox proportional hazards regression was used to estimate the hazard ratios (HRs) for the development of infertility, adjusting for age, sex, and co-morbidities. The Kaplan–Meier method was used to plot the cumulative incidence curves. We also performed negative controls to test for possible unmeasured confounding. Results. The HPV cohort had a higher risk of infertility. The adjusted HR (aHR) was found to be 1.39 (95% CI = 1.19–1.63) after adjusting for demographic characteristics and relevant co-morbidities. In the age subgroup analysis, patients with an HPV infection had an increased risk of infertility compared to the non-HPV cohort in the group aged 26 to 35 years (aHR, 1.53; 95% CI = 1.24–1.88). As we used propensity score matching to treat measurable confounders and negative controls to access unmeasured confounders, the findings of the study are robust. Conclusions. Among females of reproductive age, HPV infection is a potential risk factor that predisposes individuals to subsequent infertility.

## 1. Introduction

Infertility is defined as the inability to conceive after one year of regular intercourse without contraceptive measures involving both a male and a female partner [1]. It is a complicated disorder affecting 8%–12% of reproductive-aged couples—estimated to be more than 48 million couples worldwide—which may lead to emotional and psychological distress [2,3]. The most common identifiable causes of female infertility are ovulatory disorders, endometriosis, pelvic adhesion, tubal abnormality, uterine abnormalities, hyperprolactinemia, and endometrial lesions [3,4,5,6]. Psychological stress and lifestyle factors may influence physical responses and also play a role in infertility [7]. As a matter of fact, around 20%–60% of cases of female infertility are related to sexually transmitted infections (STIs), especially those influencing tubal–pelvic conditions; *Chlamydia trachomatis* and *Neisseria gonorrhoeae* are pathogens that have been shown to cause cervical, tubal, and peritoneal damage to the host, causing infertility [8].

Human papillomaviruses (HPVs) are double-stranded DNA viruses that only infect humans and are considered some of the most common sexually transmitted viruses [9]. HPV infection is a major cause of anogenital warts [10] and is highly related to infection-related precancerous and cancerous lesions of the cervix uteri, vulva, vagina, anus, oropharynx, and penis [11,12,13,14,15]. Other studies have found HPVs in endometrial and ovarian lesions and tissues [16,17,18]. Persistent HPV infection has been linked to chronic inflammation [19], and during the HPV life cycle, the infectious virion-producing pathway may weaken the cells it resides in, which has been linked to infertility and early abortion [20].

In this study, we hypothesized that HPV infection might be a risk factor for female infertility. We conducted this original longitudinal nationwide cohort study to explore this important issue.

## 2. Material and Methods

Data source: The Taiwan’s National Health Insurance Research Database (NHIRD) contains a large amount of data about the health of Taiwan’s residents. The NHIRD was established in 1995 when a National Health Insurance (NHI) program was launched by the Taiwanese government. In this retrospective cohort study, we utilized a sub dataset of the NHIRD, the Longitudinal Health Insurance Database (LHID), which contains the medical claim data of one million insured people from 1996 to 2013. The disease codes in the LHID were determined according to the International Classification of Disease, Ninth Revision, Clinical Modification (ICD-9-CM). The Institutional Review Board of the China Medical University in Taiwan approved this study (CMUH-104-REC2-115). Study population: To ensure the accuracy of the participants’ data, patients with missing demographic information were excluded from the analysis before the index date. All diagnosis codes in the NHIRD were recorded based upon the International Classification of Disease, Ninth Revision, Clinical Modification (ICD-9-CM). For this cohort study, we recruited patients who had been diagnosed with human papillomavirus (HPV) (ICD-9-CM code 079.4, 078.1, 795.05, 795.09, 795.15, 795.19, 796.75, or 796.79) between 2000 and 2012 as the study cohort and patients without a history of HPV as the control cohort. To ascertain that the enrollees had accurate diagnoses, the subjects in the study cohort were only confirmed if they had undergone HPV-related examinations and procedures such as colposcopy, cervical conization, or cryotherapy in the three months prior to or after the diagnosis of HPV. The index date was defined as the first date of HPV diagnosis for the case patients, and the controls were assigned a date between 2000 and 2012. To exclude the dated diagnosis issue, for each patient, the diagnosis of HPV was required to be after 2000, and the diagnosis of infertility had to be after the index date. We excluded patients who were under fifteen or above forty-five years of age, had received a hysterectomy, had been diagnosed with cancer, or had been diagnosed with infertility before the index date. Each control patient was matched with an HPV patient according to their propensity score. Patients who were not matched in the HPV cohort were excluded. The matched non-HPV patients had the same index dates and distribution of relevant co-morbidities as their paired HPV individuals. We identified the main outcome from the index date to the end of our study (31 December 2013).

## 3. Main Outcome and Co-Morbidities

The primary focus of this study was infertility (ICD-9-CM code 628). All participants were followed up until December 2013, unless they were diagnosed with infertility, withdrew from the NHI program, or died. Co-morbidities associated with a risk of infertility include endometriosis (ICD-9-CM code 617), polycystic ovarian syndrome (PCOS) (ICD-9-CM code 256.4), benign neoplasm of the ovary (ICD-9-CM code 220), pelvic inflammatory disease (PID) (ICD-9-CM code 614), uterine leiomyoma (ICD-9-CM code 218), and infertility-associated operations, such as myomectomy (ICD-9-CM procedure code 68) and tuboplasty (ICD-9-CM procedure code V26.0). Information on co-morbid medical disorders was obtained by tracing all of the ambulatory medical care and inpatient records in the NHIRD within the two years prior to the index date.

## 4. Statistical Analysis

First, the demographic characteristics, including the distributions of categorical age, sex, and co-morbidities between the HPV cohort and non-HPV comparison cohort were analyzed by chi-square tests. The incidence density of infertility per 1000 person-years was calculated in both cohorts.

Second, to investigate the possible effect of HPV, a Cox proportional hazards regression analysis was conducted to estimate the hazard ratios (HRs) and 95% confidence intervals (CIs) after the adjustment of covariates. The covariates included in the multivariable models included age, sex, co-morbidity of endometriosis, polycystic ovarian syndrome, benign neoplasm of the ovary, pelvic inflammatory disease, uterine leiomyoma, myomectomy, and tuboplasty. Propensity score matching was estimated using logistic regression to minimize the measurable confounders and potential selection bias in terms of previously mentioned variables.

Third, the Kaplan–Meier method was used to describe the cumulative incidence of infertility in the two cohorts. The difference between the two cohorts was examined by a log-rank test. The incidence of infertility was estimated by dividing the number of infertility events by the follow-up person-years for both cohorts. Figure 1 shows the Kaplan–Meier curves of the incidence of infertility in individuals with and without an HPV infection.

Fourth, to examine the effects of sex, age, and each mentioned co-morbidity on the incidence of infertility among patients with an HPV infection, we used the multivariable Cox regression model adjusted for age, sex, and co-morbidities. The HR adjusted for covariates was calculated for male and female patients and for the following age groups: 15–25, 26–35, and 36–45 years. We conducted a test to assess interactions. If the *p* value for the interaction test was statistically significant, this indicated an observable subgroup effect on the incidence rate of infertility.

Last, because lifestyle and behaviors are also important risk factors for infertility, we used negative controls to test for the possibility of such unmeasured confounding. We chose lung cancer (as falsification end point 1) and hepatocellular carcinoma (as falsification end point 2). The literature review showed that there seems to be no relationship between HPV infection and lung or hepatocellular carcinomas. However, if the result showed a significant association between HPV infection and lung cancer using the model in this study, it would indicate the presence of an unmeasured confounder (such as tobacco use) from this point of view. Again, if the result showed a positive association between HPV infection and hepatocellular carcinoma, it would indicate the presence of an unmeasured confounder (such as alcohol use) from HPV to infertility. In contrast, if the results of the negative controls showed insignificant associations between HPV infection and both malignancies, the finding of HPV being a potential risk factor for infertility would be more reliable.

## 5. Results

This study included 11,198 patients with HPV and the same number of controls. Table 1 shows the demographic and clinical variables of the HPV and control patients. Among all the participants, 41% were aged 15–25 years and the mean age was 28.5 years. The distribution of co-morbidities was similar in the two cohorts.

As shown in Figure 1, the cumulative incidence of infertility for HPV patients was significantly higher than that of patients without an HPV infection. Table 2 shows that the incidence rate of infertility in patients with HPV was 5.89 per 1000 person-years, which was higher than that of non-HPV patients (4.20 per 1000 person-years). The adjusted hazard ratio of infertility in HPV patients relative to the control patients was 1.39 (95% CI = 1.19, 1.63). Compared to patients aged 15–25 years old, patients aged 26–35 years old had a higher risk of infertility (adjusted HR (aHR) = 2.07, 95% CI = 1.76, 2.45), but patients aged 36–45 years old had a reduced risk of infertility (aHR = 0.33, 95% CI = 0.25, 0.45). The risk of infertility was higher in patients with endometriosis (aHR = 1.85, 95% CI = 1.33, 2.58), PCOS (aHR = 2.29, 95% CI = 1.54, 3.39), benign neoplasm of the ovary (aHR = 1.64, 95% CI= 1.23, 2.2), and PID (aHR = 1.62, 95% CI = 1.01, 2.6). Endometriosis, PCOS, benign neoplasm of the ovary, PID, and uterine leiomyoma are risk factors for infertility.

Appendix A shows the results of the stratification analysis for the association between HPV and infertility. In patients aged 26–35 years, the risk of infertility for HPV was 1.53 times (95% CI = 1.24–1.88) higher than that of those without a history of HPV infection. In the co-morbidity subgroup analysis, HPV infection was found to have a stronger association in patients without the co-morbidities mentioned in the study.

## 6. Negative Control Outcomes Analysis

Between 2000 and 2012, we identified 13,953 female patients from the LHID with an HPV infection. In the analysis with propensity score matching, we performed multivariable-adjusted analyses using the age-, sex-, and index date-matched controls to estimate the HRs of lung cancer and hepatocellular carcinoma in patients with an HPV infection in comparison with the non-HPV cohort. The negative control outcome showed non-significant associations between HPV infection and lung cancer (aHR, 0.85; 95% CI, 0.63–1.12) and between HPV and hepatocellular carcinoma (aHR, 0.92; 95% CI, 0.63–1.34).

We adopted this method to show that the potential magnitude of any uncontrolled confounder is minimal. However, we did not completely control for confounders in our study.

## 7. Discussion

This first retrospective cohort study using 13 years of nationwide population-based data suggests that people with HPV have a nearly 1.4-fold increased risk of developing female infertility compared with the general population. HPV is one of the most common STIs [9] and the majority of HPV infections are cleared by the host within a short period of time [21]. Clearance of an HPV infection is more difficult for older women, women with multiple infections, and those infected by high-risk HPV (HR-HPV) [22]. In our study, the stratified analysis showed that the effects of HPV infection are more significant in females aged from 26 to 35 years than in those aged from 15 to 25 years. This may be due to having an increased number of accumulated sexual partners and a lower HPV clearance ability as their age increases [22,23]. However, when comparing the age group 36–45 years to the younger age groups, the age group 36–45 years had a lower risk of infertility. This may result from the relatively lower ratio of the population aged 36–45 years that seeks infertility treatment, since older women are more likely to be married or in stable partnerships, or perhaps they previously learned of their infertility and either treated it or gave up when they were younger (in the 26–35 range).

Though studies have established an association between HPV infection and both male infertility [24,25,26,27] and assisted reproductive technology (ART) outcomes [28,29,30], the mechanism by which HPV infection decreases female fertility remains unclear. However, Yuan et al. performed a meta-analysis on HPV infection and female infertility which showed a significant association between HR-HPV and female infertility (odds ratio 2.33, 95% CI 1.42–3.83, *p* = 0.0008) but surmised that HR-HPV is more likely a trigger factor rather than an independent cause [31]. Shannon et al. found that HPV clearance was associated with increased endocervical Langerhans cells, and HPV infection was related to changes in the cervico-vaginal microbiome that correlated most closely with bacterial vaginosis (BV) [32]. Other studies have demonstrated that HPV-infected cells may actively promote chronic stromal inflammation [19,33]. BV and chronic inflammation may destruct the balance of the immune system required for optimal fertility [34,35].

HPV infection often co-exists with other conditions that are related to infertility, such as *Chlamydia trachomatis* infection and endometriosis [36,37]. We performed propensity score matching (PSM) with the same percentages of endometriosis, polycystic ovarian syndrome (PCOS), benign neoplasm of the ovary, pelvic inflammatory disease (PID), uterine leiomyoma, and infertility-associated operations in both the HPV and non-HPV groups to lower the impact of these confounding factors. Thus, the selection bias was limited. Besides, the results showed that the co-morbidities listed in Table 2 have a high risk for infertility and thus indicated that our study model was robust. Furthermore, a subgroup analysis showed that, in patients without co-morbidities associated with infertility, HPV infection had a stronger association with subsequent infertility (please refer to Appendix A). With previous HPV infection, we found that patients without endometriosis had an adjusted hazard ratio (aHR) of 1.43 (95% CI, 1.22–1.68); patients without PCOS had an aHR of 1.44 (95% CI, 1.22–1.68); patients without benign neoplasm of the ovary had an aHR of 1.42 (95% CI, 1.21–1.67); patients without PID had an aHR of 1.41 (95% CI, 1.2–1.65); patients without uterine leiomyoma had an aHR of 1.42 (95% CI, 1.21–1.66); patients without infertility-associated operation had an aHR of 1.41 (95% CI, 1.2–1.65). Thus, we concluded that HPV infection is associated with the development of infertility, especially in people without clinically relevant co-morbidities. Zhang et al. revealed that treatment and clearance of an HPV infection from the cervix improves fertility [38]. With the advent of the HPV vaccine and a massive uptake in injections, a reduction in the viral prevalence is expected, and we are looking forward to the subsequent effects on the fertility and general health of both females and males.

The strength of our study was the use of nationwide population-based data and PSM to evaluate infertility risk in patients with an HPV infection [39]. The advantages of using the NHIRD in the research include its enormous sample size, population-based data, and long-term comprehensive follow-up period. However, several limitations inherent to the use of insurance claims databases must be taken into account. First, ICD-9-CM codes for the diagnoses of HPV infection and infertility were based on administrative claims data recorded by physicians and hospitals rather than a prospective clinical setting. Inaccuracies may have resulted in misclassification, despite the fact that the Bureau of the NHI uses an auditing mechanism to minimize diagnostic uncertainty and misclassification. Some HPV-infected patients with mild symptoms do not search for medical services and they may have been included in the control cohort. However, if HPV is associated causally with the development of infertility, misclassification would bias the estimated HRs toward the null. Therefore, our findings are reliable. Second, many demographic variables were not available in the database, such as body mass index, tobacco or alcohol consumption status, lifestyle factors, socioeconomic status, family medical history, fertility status, infertile male partners, and the possibility of being influenced by HPV-infected fertile male partners. Lyu et al. showed that the overall prevalence of HPV DNA in semen was higher in fertility clinic attendees than in the general population [27], and Depuydt et al. found that couples undergoing intrauterine insemination had a significant decrease in clinical pregnancy rate when infectious HPV virions were detected in the sperm [40]. These factors contribute to the development of infertility, and these confounders are an inherent limit of the NHIRD. In the study, we applied a negative control outcome analysis to test the unmeasured confounders (lung cancer was the proxy of the effect of tobacco use and hepatocellular carcinoma the proxy of the effect of alcohol consumption). These observations demonstrated that unmeasured confounding was unlikely to have had an effect from this point of view. Third, infertility patients are more likely to receive more medical examinations and surveys that may lead to a higher detection rate for infertility-related diseases, such as endometriosis and PCOS. Fourth, an HPV vaccine has been available in Taiwan since 2006, and with more and more people receiving HPV vaccination, it may exert some protective effects against HPV infection that should be considered. However, between 2000 and 2012, the HPV vaccine was not covered by Taiwan National Health Insurance; therefore, the data are absent. Despite these limitations, our study was based on a nationwide, population-based database that included nearly all of Taiwan’s residents. The large sample size in our study contributed to its substantial statistical power and revealed an obvious association between HPV and infertility with minimal selection biases.

## 8. Conclusions

This population-based cohort study demonstrates a higher risk of female infertility in patients with previous HPV infection, especially those aged 26–35 years. Further studies are needed to clarify the underlying biological mechanisms of these associations.

## Figures and Tables

**Figure 1 ijerph-17-06505-f001:**
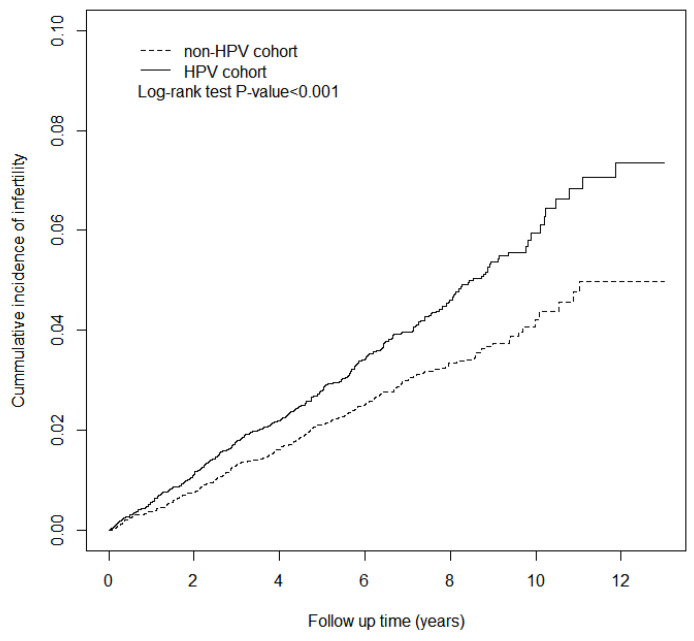
The cumulative incidence of infertility in the human papillomavirus (HPV) cohort and the non-HPV cohort.

**Table 1 ijerph-17-06505-t001:** Baseline characteristic of patients with and without HPV.

	Human Papillomavirus				
	No (N = 11,198)		Yes (N = 11,198)		
Variables	n	%	n	%	*p*-Value
Age, years					0.97
15–25	4578	41%	4589	41%	
26–35	3564	32%	3548	32%	
36–45	3056	27%	3061	27%	
mean, (SD)	28.5	−8.62	28.5	−8.6	0.93
Comorbidities					
endometriosis	403	4%	419	4%	0.59
PCOS	189	2%	202	2%	0.54
Benign neoplasm of ovary	533	5%	534	5%	1
PID	190	2%	201	2%	0.61
Uterine leiomyoma	438	4%	466	4%	0.36
infertility-associated operation	375	3%	396	4%	0.46

PCOS: polycystic ovarian syndrome; PID: pelvic inflammatory disease; n: number of patients.

**Table 2 ijerph-17-06505-t002:** Incidence rate and hazard ratio of Infertility.

	Infertility						
Variables	n	PY	IR	cHR	(95% CI)	aHR ^†^	(95% CI)
non-HPV	273	64,970	4.2	1	-	1	-
HPV	383	65,033	5.89	1.4	(1.2,1.64) ***	1.39	(1.19,1.63) ***
Age, years							
15–25	226	53,688	4.21	1	-	1	-
26–35	375	40,357	9.29	2.21	(1.88,2.61) ***	2.07	(1.76,2.45) ***
36–45	55	35,957	1.53	0.36	(0.27,0.49) ***	0.33	(0.25,0.45) ***
Comorbidities							
endometriosis							
No	611	125,611	4.86	1	-	1	-
Yes	45	4391	10.25	2.11	(1.56,2.85) ***	1.85	(1.33,2.58) ***
PCOS							
No	629	128,220	4.91	1	-	1	-
Yes	27	1782	15.15	3.11	(2.11,4.57) ***	2.29	(1.54,3.39) ***
Benign neoplasm of ovary							
No	598	124,239	4.81	1	-	1	-
Yes	58	5764	10.06	2.09	(1.6,2.74) ***	1.64	(1.23,2.2) ***
PID							
No	638	127,985	4.98	1	-	1	-
Yes	18	2018	8.92	1.8	(1.12,2.87) *	1.62	(1.01,2.6) *
Uterine leiomyoma							
No	627	125,037	5.01	1	-		
Yes	29	4966	5.84	1.17	(0.8,1.69)		
Infertility-associated operation							
No	637	125,664	5.07	1	-		
Yes	19	4338	4.38	0.86	(0.55,1.36)		

*: *p*-value < 0.05; ***: *p*-value < 0.001; PCOS: polycystic ovarian syndrome; PID: pelvic inflammatory disease. n: number of patients; PY: person-year; IR: incidence rate per 1000 person-years. cHR: crude hazard ratio; aHR: adjusted hazard ratio; †: multivariable model with HPV, age, endometriosis, PCOS, benign neoplasm of ovary, and PID.

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
