# Peer review of "Female Human Papillomavirus Infection Associated with Increased Risk of Infertility: A Nationwide Population-Based Cohort Study"

_ijerph, 2020, doi:10.3390/ijerph17186505_

Round 1

Reviewer 1 Report

I agree with conclusion that the use of population-based data and long-term comprehensive follow-up period clearly revealed higher risk of female infertility in patients with previous HPV infection.  

Some additional citations including the study related with higher risk of infertility due to the lower semen quality that significantly reduces success of the artificial insemination in the couples infected with HPV in comparison to non-infected couples could be mentioned somewhere in the introduction or discussion section.

What types of HPV viruses (all known types or just some most highly oncogenic, for example) were taken into account and were considered as causative agents of the higher risk of infertility? 

The meaning of "n" isn't explained in no one of the tables.

The title of Table 4 should be included.

Results, line 156: insert "higher" after ...1.53 times...

Author Response

Point 1: Some additional citations including the study related with higher risk of infertility due to the lower semen quality that significantly reduces success of the artificial insemination in the couples infected with HPV in comparison to non-infected couples could be mentioned somewhere in the introduction or discussion section.

Response 1: We thank the reviewer for providing the worthwhile suggestion on improving the quality of our article. We have added the content to the part of discussion. ( In the new clean version, page 7, line 230-233).

line 230-233: …Lyu et al. showed that the overall prevalence of HPV DNA in semen was higher in fertility clinic attendees than in general population and Depuydt et al. found that couples undergoing intrauterine insemination had a significant decrease in clinical pregnancy rate when infectious HPV virions were detected in the sperm.

Point 2: What types of HPV viruses (all known types or just some most highly oncogenic, for example) were taken into account and were considered as causative agents of the higher risk of infertility?

Response 2: We recruited patients who had been diagnosed with human papillomavirus (HPV) by using International Classification of Disease, Ninth. Revision, Clinical Modification (ICD-9-CM code 079.4, 078.1, 795.05, 795.09, 795.15, 795.19, 796.75, or 796.79), inclusive of low-risk (viral warts) and high-risk oncogenic types. We cannot differentiate which types of HPV are the main cause of infertility in this study.

Point 3: The meaning of "n" isn't explained in no one of the tables.

Response 3: Thank you for the reminding. We have added annotation of "n: number of patients" to all the tables with "n".

Point 4: The title of Table 4 should be included.

Response 4: We thank you for the reviewer’s suggestion. However, due to recurrent or re-infection of HPV often happens and such condition is not estimated in this study, we humbly request the reviewer to let us remove Table 4, to make the context more clear and concise.

Point 5: Results, line 156: insert "higher" after ...1.53 times...

Response 5: We thank you for the reminding and "higher than" is added.

Reviewer 2 Report

The concept and the data set are valuable.

 It is of note that detecting an STI such as HPV does itself mean of increased risk of having other STIs. Thus, HPV can easily be a surrogate marker for PID, endometriosis, need for infertility associated operations

There are concerns on data analysis and interpretation:

Table 2 and Table 3 together are confusingly redundant.

The value of Table 3 is questionable since it is based on stratification which can mask weak associations. It can be biassed by the surrogate marker nature of HPV i.e. no wonder for the lack HPV associations in Table 3. Authors should consider to miss this table.

Since the major marker of this study is the HPV infection, Table 2 adjustments should include HPV, too. It will probably eliminate the yet unexplained low hazard (i.e. protection) in age group 35-45.

The data in Table 4 for are not translated to pathomechanism. Either explicate or miss this topic.

Author Response

Point 1: It is of note that detecting an STI such as HPV does itself mean of increased risk of having other STIs. Thus, HPV can easily be a surrogate marker for PID, endometriosis, need for infertility associated operations.

There are concerns on data analysis and interpretation:

Table 2 and Table 3 together are confusingly redundant.

The value of Table 3 is questionable since it is based on stratification which can mask weak associations. It can be biased by the surrogate marker nature of HPV i.e. no wonder for the lack HPV associations in Table 3. Authors should consider to miss this table.

Response 1: We thank the reviewer for providing the worthwhile suggestion on improving the quality of our article. The purpose of Table 3 was to show that in people without comorbidities, HPV plays a role in infertility. However, it seemed redundant together with Table 2. To make our article more concise, we decided to remove Table 3 and add it to supplementary file.

Point 2: Since the major marker of this study is the HPV infection, Table 2 adjustments should include HPV, too. It will probably eliminate the yet unexplained low hazard (i.e. protection) in age group 35-45.

Response 2: The adjusted hazard ratio of infertility was obtained by including HPV, age, endometriosis, PCOS, benign neoplasm of ovary and PID into the multivariable Cox proportional model. HPV, the main variable was already included and other variables were included for adjusting. The low hazard ratio of infertility in female in 35-45 years old demonstrated that they had a lower risk to have infertility compared to female with 15-25 years old. That is reasonable.

Point 3: The data in Table 4 for are not translated to pathomechanism. Either explicate or miss this topic.

Response 3: We thank the reviewer for providing the practical suggestion. After discussion, we decided to remove Table 4.

Reviewer 3 Report

The manuscript by Hsu et al describes the risk of infertility in females with Human Papillomavirus infection.

This is an interesting study based on the information gathered from the large size of population sample analyzed of women with infertility diagnoses and corroborates previous findings by other research groups (Yuan et al, 2019 - PMID: 31987733). Statistical analyses are appropriate and comprehensive, although the findings are not particularly novel.     

The manuscript is generally well written and flows well, only requiring minor changes, for example, italicize species names and check the spacing between words. 

Figure 1 could be revised to better differentiate between the line that represents the non-HPV cohort and HPV cohorts.

Author Response

Point 1: The manuscript is generally well written and flows well, only requiring minor changes, for example, italicize species names and check the spacing between words.

Response 1: We thank you for the reviewer’s reminding and have corrected the errors.

Point 2: Figure 1 could be revised to better differentiate between the line that represents the non-HPV cohort and HPV cohorts.

Response 2: Figure 1 was modified according to the reviewer’s suggestion.

Round 2

Reviewer 2 Report

none

Author Response

All authors thank your opinion.

God speed,

Renin Chang